# Card9 Broadly Regulates Host Immunity against Experimental Pulmonary *Cryptococcus neoformans* 52D Infection

**DOI:** 10.3390/jof10060434

**Published:** 2024-06-19

**Authors:** Isabelle Angers, Wided Akik, Annie Beauchamp, Irah L. King, Larry C. Lands, Salman T. Qureshi

**Affiliations:** 1Translational Research in Respiratory Diseases Program, Research Institute of the McGill University Health Centre, McGill University, Montreal, QC H4A 3J1, Canada; isabelle.angers@mail.mcgill.ca (I.A.); wided.akik@mail.mcgill.ca (W.A.); abtsa87@gmail.com (A.B.); irah.king@mcgill.ca (I.L.K.); larry.lands@mcgill.ca (L.C.L.); 2Meakins-Christie Laboratories, Division of Experimental Medicine, McGill University, Montreal, QC H4A 3J1, Canada; 3Meakins-Christie Laboratories, Department of Microbiology and Immunology, McGill University, Montreal, QC H4A 3J1, Canada; 4Meakins-Christie Laboratories, Department of Pediatrics, McGill University, Montreal, QC H4A 3J1, Canada; 5Meakins-Christie Laboratories, Department of Medicine, McGill University, Montreal, QC H4A 3J1, Canada

**Keywords:** *Cryptococcus neoformans*, caspase recruitment domain-containing protein 9, respiratory infection, meningoencephalitis, innate and adaptive immunity

## Abstract

The ubiquitous soil-associated fungus *Cryptococcus neoformans* causes pneumonia that may progress to fatal meningitis. Recognition of fungal cell walls by C-type lectin receptors (CLRs) has been shown to trigger the host immune response. Caspase recruitment domain-containing protein 9 (Card9) is an intracellular adaptor that is downstream of several CLRs. Experimental studies have implicated Card9 in host resistance against *C. neoformans*; however, the mechanisms that are associated with susceptibility to progressive infection are not well defined. To further characterize the role of Card9 in cryptococcal infection, Card9^em1Sq^ mutant mice that lack exon 2 of the *Card9* gene on the Balb/c genetic background were created using CRISPR-Cas9 genome editing technology and intratracheally infected with *C. neoformans* 52D. Card9^em1Sq^ mice had significantly higher lung and brain fungal burdens and shorter survival after *C. neoformans* 52D infection. Susceptibility of Card9^em1Sq^ mice was associated with lower pulmonary cytokine and chemokine production, as well as reduced numbers of CD4^+^ lymphocytes, neutrophils, monocytes, and dendritic cells in the lungs. Histological analysis and intracellular cytokine staining of CD4^+^ T cells demonstrated a Th2 pattern of immunity in Card9^em1Sq^ mice. These findings demonstrate that Card9 broadly regulates the host inflammatory and immune response to experimental pulmonary infection with a moderately virulent strain of *C. neoformans*.

## 1. Introduction

*Cryptococcus neoformans* is a globally distributed soil-borne fungus that causes pneumonia following inhalation of infectious propagules. Pulmonary cryptococcosis is typically mild or even asymptomatic; however, in some cases it may progress to meningoencephalitis, a morbid condition that is universally fatal if left untreated [1]. *C. neoformans* is estimated to cause approximately 152,000 cases of meningitis annually with an estimated death toll of 112,000 [2]. Among adults living with HIV in sub-Saharan Africa, *C. neoformans* is the most common cause of meningitis and accounts for 15–20% of AIDS-related deaths. *C. neoformans* is also increasingly recognized to cause disease in patients without HIV, including solid organ transplant recipients, patients receiving immunosuppressive therapy, and individuals who are otherwise considered immunocompetent [3]. Cryptococcal disease is characterized by complex host–pathogen interactions that result in damage to both structural and immune cells and an effective crosstalk between lymphocytes and mononuclear phagocytes is required for fungal clearance [4,5].

Recognition of fungal pathogens by various families of host pattern recognition receptors (PRRs) including Toll-like receptors (TLR), C-type lectin receptors (CLRs), and intracellular Nucleotide Oligomerization Domain-like receptors (NLR) is a critical first step in activation of innate immunity against infection [6]. Upon activation, transmembrane PRRs initiate intracellular signaling cascades to trigger cellular responses such as cytokine production and phagocytosis that result in microbial killing. Members of the CLR family including Dectin-1, Dectin-2, Dectin-3, Mincle, and the Mannose Receptor have been shown to recognize various cell wall components of pathogenic fungi [7]. Following ligand recognition, CLRs recruit the intracellular Syk kinase that leads to the formation of a signaling complex consisting of the Caspase recruitment domain-containing protein 9 (Card9), B cell lymphoma 10 (Bcl10), and mucosa-associated lymphoid tissue lymphoma translocation protein 1 (Malt1) [8]. Subsequent activation of Card9 results in the induction of the canonical transcriptional activator nuclear factor kappa beta (NF-kB) and activation of mitogen-activated protein kinases (MAPK). The result of this cascade is to stimulate the production of inflammatory mediators such as Il—6, Il—12, Csf2, Tnf, and Il—1β that are crucial for initiation of antifungal immunity.

Genetic deficiency in Card9 is considered an autosomal recessive primary immunodeficiency disorder (PID) which uniquely manifests as an extreme susceptibility to the development of severe fungal infection [7,9]. To date at least 24 missense and nonsense mutations have been identified in the promotor or protein-coding region of human Card9 [10,11]. Card9 deficiency was first reported in 2009 in a consanguineous family with severe infections caused by the normal human commensal organism *Candida* sp. [12]. Card9-deficient humans may develop persistent and recurrent mucosal *Candida* sp. infections that are collectively termed chronic mucocutaneous candidiasis (CMC), and/or systemic infections that primarily manifest as fungal meningoencephalitis. *Candida* infections of the kidney, liver, or spleen that are typically affected in patients with iatrogenic immunosuppression have not been reported in Card9-deficient patients [13]. It is notable that Card9 deficiency is the only reported human genetic disorder where mucosal and systemic *Candida* disease occur together in the absence of other non-fungal infections. In humans, mucosal candidiasis control requires IL-17 signaling via Th17 cells, gamma delta T-cells and innate lymphoid cell populations [14]. Conversely, neutrophils are the most important effector cell in host defense against systemic *C. albicans* infection and neutropenia is a well-established risk factor for this disease. Interestingly, human neutrophils require Card9 for killing unopsonized yeast cells via a reactive oxygen species-independent PI3Kγ-dependent pathway, while neutrophil-mediated killing of opsonized *C. albicans* is mediated by FcRg and PKCd and is largely independent of Card9 [15]. Thus, it appears that Card9 deficiency impairs both lymphoid and myeloid host defenses that mediate different mechanisms of protective immunity at mucosal and systemic sites, respectively [9]. Card9 deficiency has also been associated with infection by a variety of other fungi including *Trichophyton* sp., *Phialophora verrucosa*, *Exophiala* sp., *Aspergillus fumigatus*, *Corynespora cassiicola*, and others [11].

Genetic deletion of Card9 in mice confers susceptibility to infection with several fungal pathogens [7]. The role of Card9 has been most extensively studied using a *C. albicans* infection model and has confirmed the crucial role that neutrophils play in host defense [16]. *In vivo*, Card9 promotes neutrophil recruitment into fungal-infected organs via CXC chemokine production. For example, in the mouse brain where opsonization is naturally low, Card9 is required for the appropriate induction of CXC chemokines by resident macrophages (i.e., microglia) and glial cells, as well as recruited neutrophils [16]. Elegant mechanistic studies have recently demonstrated that impaired control of *C. albicans* infection in the mouse brain is due to defective neutrophil recruitment mediated by a lack of Card9-dependent Il—1b and Cxcl1 production by microglia following recognition of the fungus-secreted toxin Candidalysin [16,17].

Despite the availability of excellent mouse models that accurately recapitulate the pathogenesis of *C. neoformans* pneumonia and meningitis, a clear understanding of the role that Card9 plays in host defense against this prevalent human pathogen is relatively limited. The first report to implicate Card9 in host susceptibility to *C. neoformans* demonstrated a higher lung fungal burden at day 5 and day 14 post-infection with 10^6^ colony-forming units of *C. neoformans* 3501 in Card9-deficient mice [18]. This observation was associated with reduced recruitment of interferon gamma-producing NK and memory phenotype T cells and reduced pulmonary expression of inducible nitric oxide synthase at day 3, but not at day 7, post-infection. Card9-deficient mice also had reduced expression of Il17a and the associated transcription factor RORγτ in lung homogenates at day 7 post-infection compared to wild-type mice; however, Il—17a-deficient mice actually had a significantly lower lung fungal burden at day 14 post-infection. The consequences of Card9 deficiency on *C. neoformans* 3501 dissemination to other organs, including the brain, were not described in this report. A subsequent investigation using an avirulent *C. neoformans* LW10 vaccine strain that overexpresses the zinc finger 2 transcription factor showed that Card9 was required for protection against progressive pulmonary infection and dissemination to the spleen but not the brain. Notably, no differences in lung leukocyte recruitment to infected organs were demonstrable; however, Card9-deficient mice had decreased M1 and increased M2 lung macrophage polarization following LW10 infection. Finally, following challenge with the highly virulent *C. neoformans* H99 strain, wild-type C57BL/6 mice survived longer than Card9-deficient mice (median survival 21 days versus 26 days) but all mice succumbed by day 29 post-infection. No significant differences of fungal burden in the lungs, brain, or spleen were observed between wild-type and Card9-deficient mice in this model, nor were there any identifiable differences in lung leukocyte recruitment, anticryptococcal activity of macrophages or dendritic cells, or macrophage polarization [19].

Taken together, these findings strongly suggest that Card9 regulates various aspects of host immunity against pulmonary cryptococcal infection such as lung macrophage polarization. Conversely, these results also raise several questions about the underlying mechanisms of disease, including an explanation for the lack of differential lung leukocyte recruitment, despite selectively higher chemokine expression (*Ccl2*, *Ccl3*, *Ccl11*) and a higher LW10 fungal burden in Card9-deficient mice. In addition, the absence of differential fungal growth in the brain despite significant differences in survival of wild-type and Card9-deficient mice following LW10 or H99 infection is unexplained. Integration of these findings into a coherent model is limited by the use of diverse cryptococcal isolates, including a laboratory strain (B3501), an avirulent strain (LW10), or a highly virulent strain (H99). Finally, all of these studies were performed in the naturally susceptible C57BL/6 inbred mice that developed progressive *C. neoformans* infection in the presence of a functional Card9 gene [20,21]. Thus, to gain further insight into the role of Card9 during the pathogenesis of cryptococcal pneumonia we challenged wild-type and Card9-mutant mice on the Balb/c background with a moderately virulent serotype D strain of *C. neoformans* (52D) using well-established experimental conditions and analyzed host resistance and immune responses at serial time points up to day 28 post-infection. Our data demonstrate that Card9 is a crucial regulator of pulmonary fungal burden that is associated with differential expression of lung inflammatory mediators, immune cell recruitment, and T cell polarization, as well as fungal dissemination and growth in the brain, and survival.

## 2. Materials and Methods

Balb/c mice were purchased from Charles River Laboratories and subsequently bred and maintained in our specific-pathogen-free (SPF) facility. To generate mutant mice, a gRNA (TCTACTACCCTCAGTTATAC) designed to target exon 2 of the Card9 gene was purchased from Integrated DNA Technologies Inc. (IDT). The gRNA and Cas9 protein were complexed and microinjected together into the pronucleus of Balb/c embryos at the McGill Integrated Core for Animal Modeling (MICAM). These embryos were then transferred into pseudo-pregnant CD-1 females (Charles River Laboratories, Saint-Constant, QC, Canada) to generate potential F0 mice. Live pups were genotyped by PCR amplification of genomic DNA extracted from ear biopsies. Primers flanking exon 2 of the gene Card9 were used for the genotyping: Card9-fwd (GCAGGGCGCCTTATTCAATG) and Card9-rev (GGCTCCCCTTCTAGAGACCA). The PCR products were visualized on a 1% agarose gel and the products were sent for Sanger sequencing (Génome Québec, Montréal, QC, Canada). A founder male was found to have a 177-base pair (bp) deletion in one allele of the Card9 gene that includes part of intron 1–2 and part of exon 2. This mutation was germline transmissible and was named Card9^em1Sq^. The male founder was outcrossed to wild-type Balb/c females and heterozygous Card9^em1Sq^ F1 mice were subsequently intercrossed to generate homozygous Card9^em1Sq^ mice. All animals were maintained in compliance with the Canadian Council on Animal Care, and all experiments were approved by the McGill University animal care and use committee. All experimental groups described in this report include an equal number of male and female mice.

*C. neoformans* 52D (ATCC 24067) was grown and maintained on Sabouraud dextrose agar (SDA) (BD, Becton Dickinson and Company, Mississauga, ON, Canada). To prepare an infectious dose, a single colony was suspended in Sabouraud dextrose broth (BD) and grown to early stationary phase (48 h) at room temperature on a rotator. The stationary phase culture was then washed with sterile phosphate-buffered saline (PBS) and diluted to 2 × 10^5^ CFU per ml in sterile PBS. The fungal concentration of the experimental dose was confirmed by plating a dilution of the inoculum on SDA and counting the CFU after 72 h of incubation at room temperature.

For intratracheal administration of *C. neoformans*, 8-week-old mice were anesthetized with 128 mg/kg of ketamine (Vetoquinol) and 6.4 mg/kg of xylazine (Elanco, Mississauga, ON, Canada) subcutaneously. Briefly, a 30½ needle mounted on a 1 mL syringe containing 2 × 10^5^ CFU *C. neoformans*/mL was inserted into the trachea under direct vision and 50 mL of inoculum followed by 50 mL of air was dispensed into the lungs. The needle was removed immediately after injection and the incision was closed using coated vicryl 5–0 resorbable suture (Ethicon, Markham, ON, Canada). The mice were monitored daily following surgery.

After mice were euthanized with CO_2_, their infected lungs and brain were excised and placed in sterile, ice-cold PBS. Tissues were then homogenized using a glass tube and pestle attached to a mechanical tissue homogenizer (Glas-Col, Terre Haute, IN, USA), and plated at various dilutions on Sabouraud dextrose agar. Plates were incubated at room temperature for 72 h, and CFU were counted.

Following euthanasia, lungs were perfused with ice-cold PBS via the right ventricle of the heart. Using 10% neutral buffered formalin acetate (Sigma, Oakville, ON, Canada), the lungs were inflated to a pressure of 25 cm H_2_O and fixed overnight. Subsequently lungs were embedded in paraffin, sectioned at 5 μm, and stained with hematoxylin eosin (H&E), periodic acid-Schiff (PAS), or mucicarmine reagents at the Histology Facility of the Goodman Cancer Research Centre (McGill University, Montreal, QC, Canada). Representative photographs of lung sections were acquired with a ZEISS Axio Imager M2 microscope and ZEISS Zen 3.3 (blue edition) software (Zeiss, Dorval, QC, Canada).

For analysis of total lung cytokine and chemokine production, mice were euthanized and lungs flushed with 10 mL of ice-cold PBS. Whole lungs were homogenized in 2 mL PBS with Halt protease and phosphatase inhibitor cocktail (Thermo Scientific, Mississauga, ON, Canada) using a sterilized glass tube and pestle attached to a mechanical tissue homogenizer (Glas-Col, Terre Haute, IN, USA) and spun at 12,000 rpm for 20 min. Supernatants were collected, and aliquots were stored at −80 °C for further analysis. Total protein concentration of each sample was measured using Pierce BCA Protein Assay kit (Thermo Scientific). The whole-lung level of IFN-γ (DY485) was analyzed using a DuoSet enzyme-linked immunosorbent assay (ELISA) kit (R&D Systems, Minneapolis, MN, USA). Levels of Kc/Cxcl1, Lif, Lix/Cxcl5, Mip-2/Cxcl2, Mip-1α/Ccl3, Eotaxin/Ccl11, G-Csf, Mcp-1/Ccl2, Vegf-a, Il—4, Il—5, Il—6, Il—1β, Il—10, Il—13, Il—17A and Tnf-α were quantified by multiplex ELISA using the MILLIPLEX Mouse Cytokine/Chemokine Magnetic Bead Panel (MCYTOMAG-70K, MilliporeSigma, Oakville, ON, Canada) according to the manufacturer’s protocol. Multiplex ELISA data were acquired with Luminex MAGPIX instrument and xPONENT platform, and analysis was performed on MILLIPLEX Analyst software (version 5).

To perform flow cytometry, the lungs and trachea were excised using sterile technique after flushing the blood with 10 mL of sterile PBS. Lungs were transferred to a sterile Petri dish and the lobes were injected with 5 mL of enzyme cocktail composed of 2 mg/mL Collagenase D (Roche, Laval, QC, Canada) and 80 U/mL DNase I (Roche, Laval, QC, Canada) in HBSS (Wisent, Saint-Jean-Baptiste, QC, Canada) through the trachea. Lung tissue (without the trachea) and any extra enzyme cocktail was transferred to a gentleMACs C tube (Miltenyi, Gaithersburg, MD, USA) and kept on ice until processing. Subsequently, a single cell suspension was obtained using the gentleMACS Octo Dissociator with heaters (Miltenyi, Gaithersburg, MD, USA) and its program 37C_m_LDK_1. Following dissociation, single cell suspensions were filtered through a 100-micrometre and 40-micrometre cell strainers (BD). Red blood cells were removed using 1× RBC lysis buffer (BioLegend, San Diego, CA, USA) before the cells were counted. Lung cells (3 × 10^6^ cells) were stained with fixable viability dye eFluor780 (eBioscience [eBio], Mississauga, ON, Canada) at the concentration of 1:10,000 for 30 min (4 °C). The cells were then washed with PBS supplemented with 0.5% BSA (Wisent, Saint-Jean-Baptiste, QC, Canada) and incubated with anti-CD16/32 (93, eBio) at a concentration of 1:100 in PBS/0.5% BSA at 4 °C for 10 min. Single-cell suspensions were subsequently stained at 4 °C for 30 min with either an adaptive or innate immune system panel composed of fluorescence-conjugated anti-mouse monoclonal antibodies purchased from eBio, BD, and BioLegend (BL). Adaptive immune system panel: CD45-BV506 (30-F11; eBio), CD3e-PE-Cy7 (17A2; eBio), CD4-V421 (GK1.5; BL), CD8-PerCP-eF710 (53—6.7; eBio). Innate immune system panel: CD45-PE-Cy7 (30-F11; BL), CD11b-BV785 (M1/70; BL), CD11c-APC (N418; eBio), Ly-6G-BV421 (1A8; BL), Ly-6C-BV711(HK1.4; BL), F4/80-BUV395 (T45—2342; BD), SiglecF-PerCP-eF710 (1RNM44N; eBio), and NK1.1 (PK136; BL). All cells were subsequently washed with PBS/0.5% BSA and resuspended in 1% paraformaldehyde. Cells were acquired on BD flow cytometers (Canto II or Fortessa X-20) with FACSDiva Software (version 8.0.3). Analyses were performed using FlowJo software v.10.1 (TreeStar, Ashland, OR, USA). Gating was conducted using Fluorescence Minus One (FMO) controls.

For intracellular cytokine staining of T cells, lungs were processed as described above. Cells (5 × 10^6^ cells) were plated and stimulated for 3 h with 1× of Cell Activation Cocktail (with Brefeldin A) (BL) that contains PMA (phorbol 12-myristate-13-acetate), ionomycin, and Brefeldin A. A control stimulation was also carried out without the presence of PMA/ionomycin in PBS but in the presence of brefeldin A (GolgiPlug; BD) for 3 h. Cells were then washed, blocked with anti-CD16/32 antibodies (93; eBio), and stained with a surface antibody cocktail consisting of the fluorescence-conjugated anti-mouse monoclonal antibodies CD3-BUV737 (17A2; eBio), CD4-FITC (RM4—5; BL), CD8-PerCP-eF710 (H35—17.2; eBio), and CD45-BUV395 (30-F11; BD). The cells were then fixed, permeabilized, and stained with IL-13-PE-Cyanine7 (eBio13A, eBio), IFN-g-APC (XMG1.2; BL), and IL-17A-PE (eBio1787; eBio). Cells were acquired on BD flow cytometer Fortessa X-20 with FACSDiva Software. Analyses were performed using FlowJo software v.10.1 (TreeStar). Gating was performed using Fluorescence Minus One (FMO) controls.

For statistical analysis of all experiments, the mean and standard error of the mean (SEM) are shown unless otherwise stated. To test the significance of single comparisons, an unpaired Mann–Whitney test was applied with a threshold *p* value of 0.05. A Log-rank (Mantel-Cox) test was performed to analyze the survival curves. The significance of the difference in brain dissemination rates was determined using a Fisher’s exact test. All statistical analyses were performed with GraphPad Prism software version 10.1 (GraphPad Software Inc., Toronto, ON, Canada).

## 3. Results

### 3.1. Card9^em1Sq^ Mice Have Impaired Survival and an Increased Fungal Burden in the Lung and Brain Following C. neoformans 52D Infection

To evaluate the role of the Card9-mediated signaling on host defense against *C. neoformans* infection, we used CRISPR-Cas9 to delete 177 bp of the Card9 gene in Balb/c mice that encodes amino acid 63–92 of the Card domain (Appendix A). Truncation of 30/86 amino acids of the Card domain is predicted to disrupt homotypic interactions that result in formation of a Card9–Bcl10–Malt1 signaling complex [22]. The newly generated mouse strain name was designated Card9^em1Sq^. Subsequently, to determine whether Card9-dependent immune mechanisms contribute to host resistance, we infected Card9^em1Sq^ and wild-type Balb/c mice with *C. neoformans* 52D via the intratracheal route and analyzed survival. Throughout the observation period no deaths were observed in wild-type mice; however, Card9^em1Sq^ mice started to die at 20 days post-infection and had a 50% and 100% mortality rate at 29 and 36 days post-infection, respectively (Figure 1A). To ascertain whether mortality was associated with a failure to control the growth of *C. neoformans* 52D we quantified lung and brain fungal burden in Card9^em1Sq^ and Balb/c mice at weekly intervals following intratracheal infection. A comparable lung fungal burden of 10^7^ CFU was observed at day 7 post-infection in both strains, suggesting that the Card9-dependent signaling did not have a marked effect on the initial host response to *C. neoformans*. In Card9 ^em1Sq^ mice the cryptococcal lung burden remained stable at approximately 10^7^ CFU from day 14 post-infection up to day 28 post-infection. In contrast, a significant reduction in pulmonary fungal burden was observed in Balb/c mice beginning at day 14 post-infection and was most pronounced at day 28 post-infection (Figure 1B). Remarkably, the lung CFU in Balb/c mice was more than 100-fold lower than that in Card9^em1Sq^ mice at day 28 post-infection (log_10_ CFU, 5.14 ± 0.28 versus 7.35 ± 0.06; *p* ≤ 0.000001). Collectively, these data show that activation of a Card9-dependent host responses makes a significant contribution to the control of *C. neoformans* 52D growth in mouse lungs.

As infection of the central nervous system is the most serious and potentially fatal consequence of pulmonary cryptococcal disease, we also examined whether mutation of Card9 altered the rate of dissemination and fungal burden in the brain. Analysis of CFU from infected Card9^em1Sq^ and Balb/c mice showed comparable rates of dissemination to the brain at day 7 and day 14 post-infection (Figure 1C); however, at day 21 post-infection, Card9^em1Sq^ mice demonstrated a significant increase in the rate of brain dissemination (10/13 versus 3/14; *p* ≤ 0.01) and cerebral fungal burden (log_10_ CFU, 3.82 ± 0.66 versus 1.11 ± 0.59; *p* ≤ 0.01) compared to Balb/c mice. The difference in brain dissemination was even more pronounced at day 28 post-infection (12/16 versus 1/16; *p* ≤ 0.0001), and the brain fungal burden was significantly higher (log_10_ CFU, 3.48 ± 0.61 versus 0.125 ± 0.125 (*p* ≤ 0.0001) in Card9^em1Sq^ compared to Balb/c mice. Therefore, compared to the Balb/c strain, Card9^em1Sq^ mice have a highly susceptible phenotype that is characterized by an inability to control fungal growth in the lung followed by increased dissemination and fungal growth in the brain that results in 100% mortality.

### 3.2. Pulmonary Inflammation Is Reduced and Altered in Card9^em1Sq^ Lungs Following C. neoformans 52D Infection

To characterize the effect of Card9 mutation on lung pathology after challenge with *C. neoformans*, we analyzed tissue sections from Card9^em1Sq^ and Balb/c mice at serial time points after infection (Figure 2). At day 21 post-infection, H&E staining revealed foci of intense inflammatory infiltration in Balb/c mice compared to Card9^em1Sq^ mice (Figure 2A,B). Periodic acid Schiff staining demonstrated abundant mucus secretion and goblet cell metaplasia in the airways of Card9^em1Sq^ mice (Figure 2D) but were notably absent from Balb/c airways (Figure 2C) at day 21 post-infection. Finally, mucicarmine staining of Card9^em1Sq^ lungs at day 21 post-infection revealed numerous extracellular *C. neoformans* organisms that were heavily encapsulated (Figure 2F), while only poorly encapsulated intracellular cryptococci were observed in Balb/c lungs (Figure 2E). Collectively, the lung histology at day 21 post-infection demonstrates that Card9^em1Sq^ mice have reduced leukocyte recruitment and enhanced signs of type 2 airway inflammation that are associated with a markedly greater abundance of encapsulated extracellular cryptococci compared to Balb/c mice.

### 3.3. Cytokine and Chemokine Production Is Decreased in the Lungs of Card9^em1Sq^ Mice Following C. neoformans 52D Infection

Previous studies have shown that various inbred or genetically modified mouse strains that are susceptible to progressive cryptococcal infection develop a Th2 pattern of adaptive immunity, while resistant strains mount a Th1 response that is crucial for fungal clearance [23,24,25]. Therefore, we sought to determine whether the susceptibility of Card9^em1Sq^ mice to progressive *C. neoformans* infection was associated with Th1 or Th2 immune polarization. Accordingly, we quantified the expression of proinflammatory mediators (I1—1β, Il—6, Tnf-α, G-Csf, and Vegf), Th1-type cytokines (Ifn-γ), Th2-type cytokines (Il—4, Il—5, and Il—13), Th17-type cytokines (Il—17), and chemokines (Cxc1, Cxcl2, Cxcl5, Ccl2, Ccl3, Ccl11) in total lung homogenates of Card9^em1Sq^ and Balb/c mice prior to infection and at day 14 post-infection (Figure 3). Prior to challenge with intratracheal *C. neoformans*, these mediators were expressed at a level near the limit of detection including several that were below the limit of detection (Il-6, Tnf-α, G-Csf, Cxcl5, Ccl2, Il—17, Il—4, Il—5, Il—13), and there were no significant expression differences between the two strains (Figure 3A–D). Conversely, at day 14 post-infection, the expression of all measured proinflammatory, Th1-, and Th17-associated cytokines was significantly higher in the lungs of Balb/c compared to Card9^em1Sq^ mice (Figure 3A–C), while the expression of Th2-associated cytokines was significantly higher in Card9^em1Sq^ mice (Figure 3D). All measured chemokines also showed significantly higher expression in Balb/c compared to Card9^em1Sq^ mice with the exception of Ccl2 that had a trend towards higher expression in Balb/c lungs that was not statistically significant. Taken together, these findings demonstrate that Card9 broadly regulates the expression of inflammatory, Th1-, Th2-, and Th17-associated cytokines as well as several chemokines in mouse lungs following intratracheal infection with *C. neoformans*. Specifically, Balb/c mice develop a proinflammatory and Th1-associated immune response, while Card9^em1Sq^ mice express Th2-associated cytokines.

### 3.4. Card9^em1Sq^ Mice Recruit Fewer CD4^+^ Lymphocytes to the Lung Following C. neoformans 52D Infection

Adaptive immunity that is principally mediated by CD4^+^ and CD8^+^ T lymphocytes is required for effective clearance of *C. neoformans* from the mouse lung [26,27,28]. Accordingly, we compared lymphoid cell populations in the lungs of Card9^em1Sq^ and Balb/c mice at serial time points following intratracheal *C. neoformans* infection (Figure 4). In both strains, the number of CD4^+^ cells was lowest and comparable prior to infection, and peaked at day 14, with declining numbers at day 21 and day 28 post-infection. Notably, at days 14, 21, and 28 post-infection, the number of pulmonary CD4^+^ T lymphocytes was significantly higher in Balb/c mice compared to Card9^em1Sq^ mice (Figure 4A). In both Balb/c and Card9^em1Sq^ mice, the number of lung CD8^+^ T cells was highest prior to infection and declined at days 14, 21, and 28 post-infection (Figure 4B). No differences in the number of lung CD8^+^ T cells was observed between the two strains, with the exception of a small yet statistically significant increase in Balb/c compared to Card9^em1Sq^ mice at day 14 post-infection (Figure 4B).

### 3.5. Pulmonary CD4^+^ T Cells from Card^9em1Sq^ Mice Show Diminished Th1-Associated and Increased Th2-Associated Cytokine Production in Response to C. neoformans Infection

Analysis of whole-lung lysates from Balb/c and Card9^em1Sq^ mice showed significant differences in the expression of Th1- and Th2-associated cytokines during *C. neoformans* infection at day 14 post-infection (Figure 3A–D). To determine whether altered T cell polarization could explain this difference, analysis of cytokine production by CD4^+^ cells was performed at the same time point. Single-cell suspensions from infected Card9^em1Sq^ and Balb/c lungs were restimulated with PMA and ionomycin to increase cytokine production and stained for intracellular Ifn-g, Il—13, and Il—17a. Balb/c mice had a significantly lower frequency of Il—13^+^ CD4^+^ cells, while the frequency of Ifn-γ^+^ CD4^+^ cells was higher in Balb/c compared to the Card9^em1Sq^ strain. The frequency of Il-17^+^ CD4^+^ T cells was comparable in the lungs of Balb/c and Card9^em1Sq^ mice (Figure 4C,D).

To characterize the effect of Card9 mutation on the cellular immune response to *C. neoformans*, flow cytometry analysis of whole-lung digests was performed on Card9^em1Sq^ and Balb/c mice prior to intratracheal challenge and at days 14, 21, and 28 after infection. Each of the latter three time points corresponds to a significant difference in lung fungal burden between Card9^em1Sq^ and Balb/c mice (Figure 1B). Comparison of uninfected Card9^em1Sq^ and Balb/c mice showed relatively few lung leukocytes with no significant differences between the strains (Figure 5A–H). The total number of CD45^+^ cells was greatest at day 14 post-infection and was comparable between Card9^em1Sq^ and Balb/c mice at all time points (Figure 5A). The number of alveolar macrophages was also greatest at day 14 post-infection and was comparable between strains at day 14 and day 21 (Figure 5B). Card9^em1Sq^ mice had a small but statistically significant increase in alveolar macrophage number compared to Balb/c mice at day 28 post-infection (Figure 5B). The number of lung monocytes and dendritic cells peaked in both strains at day 14 post-infection and was significantly higher in Balb/c mice compared to Card9^em1Sq^ mice at day 14, day 21, and day 28 post-infection (Figure 5C,E). The number of lung macrophages was significantly higher in Card9^em1Sq^ mice at day 21 post-infection and was comparable between strains at all other time points (Figure 5D). In both mouse strains the number of neutrophils increased from day 14 to day 21 and was markedly reduced at day 28 after *C. neoformans* infection. For all time points after infection Balb/c mice had a significantly greater number of lung neutrophils compared to Card9^em1Sq^ mice (Figure 5F). The number of eosinophils was highest at day 14 post-infection in both strains and decreased at subsequent time points. Compared to the Balb/c strain, Card9^em1Sq^ mice had a small yet statistically significant increase lung eosinophil number at day 21 and day 28 post-infection (Figure 5H). Taken together, these findings indicate that Card9-dependent processes have a significant effect on lung recruitment of monocytes, dendritic cells, and neutrophils following *C. neoformans* 52D infection in Balb/c mice. In contrast, Card9^em1Sq^ mice have significantly lower numbers of these myeloid cell subsets in the lung and are characterized by the development of mild pulmonary eosinophilia after *C. neoformans* infection.

## 4. Discussion

Infection by *Cryptococcus neoformans* begins with the inhalation of poorly encapsulated infectious propagules that are detected by cells of the innate immune system [19,29]. Lung resident alveolar macrophages and dendritic cells mediate this function through expression of various pattern recognition receptors including members of the C-type lectin receptor (CLR) family [30]. CLRs are a heterogeneous superfamily of soluble and transmembrane proteins with a characteristic C-type lectin domain that recognizes carbohydrate structures of the fungal cell walls such as glucans, mannans, and chitin. *C. neoformans* is detected by the mannose receptor (CD206) [31], Dectin-2 [32], and DC-SIGN [33] but is not recognized by Dectin-1, Dectin-3, and Mincle [34,35,36].

Many PRRs use signaling and/or adaptor proteins that are shared with other members of the same family in order to activate inflammatory/innate immune responses [37]. Detection of fungal infection by several host CLRs including Dectin-1, -2, -3 and Mincle triggers the recruitment and activation of spleen tyrosine kinase (Syk) that leads to the formation of a caspase recruitment domain-containing protein 9 (Card9)–B-cell lymphoma 10 (Bcl10)–mucosa-associated lymphoid tissue lymphoma-translocation gene 1 (Malt1) scaffold complex [38]. Formation of the Card9–Bcl10–Malt1 complex can activate NFkB and MAP kinases which result in phagocytosis, DC maturation, and induction of proinflammatory cytokines. Human *CARD9* mutations are particularly associated with susceptibility to *Candida albicans* infection of the central nervous system, al-though infection with other uncommon fungal strains has also been reported [11]. Similarly, Card9^−/−^ mice are more susceptible to *C. albicans* infection of the brain owing to defective neutrophil recruitment that is mediated via Card9-dependent production of IL-1β and CXCL1 by microglia [17].

Several groups have reported that Card9-deficient mice are susceptible to respiratory infection with various cryptococcal strains. The first report to implicate Card9 in host resistance following high dose infection with *C. neoformans* B3501 identified early and transient reductions in the recruitment of interferon gamma-producing NK and memory phenotype T cells to the lung; however, the origin, specificity, and function of the latter cell subset was not defined and extrapulmonary fungal dissemination or survival differences were not reported [18]. A subsequent study found that *C. neoformans* LW10, an attenuated and normally avirulent vaccine strain, causes a progressive and disseminated infection in Card9-deficient mice [39]. Nevertheless, an unexplained lack of differences in leukocyte recruitment at infected sites sharply contrasts with observations showing a profound Card9-dependent defect in neutrophil recruitment following *C. albicans* infection [16,17]. Finally, the shorter survival time of Card9-deficient mice following *C. neoformans* H99 infection compared to controls was not attributable to fungal burden differences in the lungs, brain, or spleen, nor was it associated with differences in lung leukocyte recruitment, macrophage or dendritic cell anticryptococcal activity, or macrophage polarization [39]. The demonstration of comparable cryptococcal burden in the brains of Card9^−/−^ and wild-type mice contrasts with the significantly increased *C. albicans* load in the brains of Card9^−/−^ mice [17]. Taken together, these observations point to a non-redundant role for Card9 in anti-cryptococcal host resistance, yet they also raise several important questions about how Card9-dependent immune mechanisms contribute to the pathogenesis of cryptococcal disease.

To further characterize the consequences of Card9 deficiency on anti-cryptococcal immune responses, we used a well-established model of intratracheal infection with 10^4^ CFU of *C. neoformans* 52D, a moderately virulent serotype D clinical isolate. We hypothesized that the effect of Card9 deficiency might be confounded by the natural susceptibility of C57BL/6 inbred mice to progressive infection with *C. neoformans* 52D [23]. Therefore, we used CRISPR-Cas9 technology to generate mice lacking exon 2 of the Card9 gene on the naturally resistant Balb/c genetic background (Card9^em1Sq^) and characterized fungal growth, survival, inflammatory mediator production, and lung leukocyte recruitment at serial time points after infection.

The first notable observation of this study was the highly susceptible phenotype of Card9^em1Sq^ mice compared to the inbred Balb/c strain. In response to infection with 10^4^ CFU of *C. neoformans* 52D, Card^9em1Sq^ mice began to die at 20 DPI, and all had succumbed by 36 DPI; conversely, none of the Balb/c mice died during the same observation period. According to the damage response framework, the mechanism of host death could be attributable to uncontrolled fungal replication caused by a lack of host immunity, or could be due an exuberant or dysregulated host inflammatory/immune response [40]. To evaluate these possibilities, fungal burden was quantified in the lungs and brain as a marker of local and disseminated infection, respectively. The pulmonary fungal burden was comparable between Card9^em1Sq^ and Balb/c mice up to 14 DPI, suggesting that the Card9 mutation does not significantly alter the innate immune response to *C. neoformans* 52D. Relative to 14 DPI, the fungal burden of Balb/c mice was significantly reduced at 21 DPI and was 100-fold lower at 28 DPI, while no reduction was observed in Card9^em1Sq^ mice, suggesting that Card9 activates adaptive immunity to effectively control lung fungal growth.

Meningitis caused by dissemination of pulmonary *C. neoformans* infection to the central nervous system causes severe morbidity and mortality. To evaluate the contribution of Card9-dependent mechanisms to containment of pulmonary cryptococcal infection the incidence of dissemination and fungal burden in the central nervous system were determined following intratracheal *C. neoformans* infection. A comparable rate of dissemination and fungal burden were observed between Balb/c and Card9^em1Sq^ mice at 7 and 14 DPI; however, at 21 and 28 DPI, Card9^em1Sq^ animals had a significantly higher incidence of central nervous system dissemination as well as a markedly higher brain fungal burden. Thus, a lack of Card9-dependent mechanisms results in greater dissemination and replication of *C. neoformans* 52D in the brain of Balb/c mice. Previous reports have shown that human and mouse Card9 deficiency is associated with spontaneous and progressive cerebral nervous system candidiasis. Elegant mechanistic studies in mice demonstrated that Il-1b and Cxcl1 production by microglia in a Card9-dependent manner is required for recruitment of neutrophils that control cerebral *C. albicans* infection. Although Card9 deficiency has not been associated with human cryptococcal meningitis, the experimental findings in the current report suggest the role of Card9 in antifungal host defense of the brain may not be limited to *Candida* sp. Accordingly, it will be of interest to investigate the precise immune defects that are regulated by Card9 in the brain in response to disseminated cryptococcal infection and to compare these to experimental cerebral candidiasis.

Numerous studies have demonstrated that a Th1 or Th2 pattern of pulmonary inflammation is associated with resistance or susceptibility to progressive *C. neoformans* infection (reviewed in [41]). Specifically, classical (M1) macrophage polarization and interferon-gamma (Th1) production by CD4^+^ T cells are two principal effector mechanisms for control and elimination of pulmonary cryptococcal infection while alternative (M2) macrophage polarization and interleukin-4 and/or interleukin-13 (Th2) production by CD4^+^ T cells are permissive [42]. Previous studies have shown that STAT1 activation within macrophages is required for M1 polarization and anti-*C. neoformans* activity via the production of nitric oxide (NO) [42,43]. To determine whether progressive cryptococcal infection in Card9-deficient mice was associated with a Th2 pattern of host response, lung histology, inflammatory mediator production, and cellular recruitment to the lung were performed. The presence of numerous heavily encapsulated *C. neoformans* in association with clear evidence of airway goblet cell metaplasia and mucus production in lung sections of Card9^em1Sq^ but not Balb/c mice is consistent with a Th2 response. The lungs of Card9^em1Sq^ mice also had significantly higher levels of Th2 cytokines and significantly lower levels of Th1 and Th17 cytokines compared to Balb/c mice. These observations suggest that Card9 may regulate M1 macrophage polarization in response to pulmonary cryptococcal infection through STAT1 activation. Characterization of STAT1 phosphorylation, M1- and M2-associated gene expression, as well as NO production, in macrophages from WT and Card9^em1Sq^ mice that have been infected with *C. neoformans* could confirm this possibility. Th17-associated cytokines, including Il17a, are predominantly produced by neutrophils during pulmonary infection and contribute to disease resolution, but are not required for cryptococcal eradication [44]. Neutrophil depletion prior to intranasal infection with *C. neoformans* H99g (engineered to produce Ifn-γ) did not alter lung fungal burden but increased the level of Il17a in association with an increased frequency of Il17a^+^ γδ T cells [45]. Notably, Card9^em1Sq^ mouse lungs had significantly lower levels of all measured inflammatory mediators and chemokines, as well as a non-statistically significant increase in *Ccl2*, indicating that Card9-dependent pathways regulate the overall intensity of the inflammatory and chemotactic response following pulmonary cryptococcal infection.

Defective cellular immunity is a major risk factor for progressive and/or disseminated cryptococcal infection and is most commonly observed with HIV infection and other immunosuppressive conditions that impair the number and/or function of CD4^+^ T lymphocytes. To determine whether Card9 deficiency alters the pattern of lung immune cell recruitment, flow cytometry was performed at serial time points after *C. neoformans* infection. Compared to the Balb/c strain, Card9^em1Sq^ mice recruited fewer CD4^+^ T lymphocytes to the lung from 14–28 DPI. Phenotypic characterization by intracellular cytokine staining showed a higher and lower frequency of Il-13^+^ CD4^+^ and Ifnγ^+^ CD4^+^ cells in Card9^em1Sq^ compared to Balb/c mice, respectively.

Myeloid-derived cells including monocytes and neutrophils are rapidly recruited to the lung following intratracheal cryptococcal infection [45,46]. Most experimental studies have shown that circulating monocytes are important to cryptococcal clearance and differentiate into monocyte-derived macrophages and dendritic cells that are capable of phagocytosing and destroying *C. neoformans* [47]. Conversely, failure to recruit these cells leads to Th2-type responses with increased lung fungal burden [46]. Compared to monocytes, macrophages, and dendritic cells, the role of neutrophils is less well understood [48]. In vitro, human and mouse neutrophils are highly effective at killing *C. neoformans* through oxidative and non-oxidative mechanisms [49,50]. In mice, neutrophils have also been shown to capture and remove cryptococci from the brain vasculature [51]; however, antibody-based depletion studies suggest that neutrophils may have a detrimental effect on the control of cryptococcal infection [52]. Card9 is mainly expressed by cells of the myeloid lineage; accordingly, recruitment of these cells was analyzed at serial time points after cryptococcal infection. Card9^em1Sq^ mice recruited significantly fewer monocytes, dendritic cells, and neutrophils to the lung following *C. neoformans* infection compared to Balb/c mice, confirming that this adaptor regulates myeloid cell responses. Notably, Card9^−/−^ mice also have markedly impaired neutrophil recruitment to the brain following *C. albicans* infection [16], suggesting that this signaling pathway is activated by both pathogens and mediates myeloid cell recruitment to a broad range of fungi. Finally, a trend towards greater lung eosinophil recruitment was observed in Card9^em1Sq^ mice and is consistent with the observed cytokine production pattern and Th2 cell polarization.

The current study builds on previous reports that have implicated Card9 in host resistance to *C. neoformans* infection. The choice of Balb/c mice was based on their naturally occurring relative resistance to *C. neoformans* and underlines the importance of host genetic background when analyzing immune responses to infection. Through a comprehensive analysis of Balb/c mice that have wild-type or mutant Card9 we have elucidated broad effects of this adaptor molecule on the innate and adaptive immune response to *C. neoformans* 52D, a moderately virulent clinical isolate that has been widely used to study pulmonary disease pathogenesis. Our data indicate that fungal recognition and signaling through Card9 activate innate and adaptive immunity in the lungs that controls dissemination and replication in the brain. Some strengths of our report include the use of CRISPR-Cas9 gene editing technology to precisely introduce a defined genetic modification in Card9 without affecting adjacent sequences, and the use of a well-established intratracheal infection model to comprehensively characterize lung inflammatory and immune responses. The present data also extend the findings from previous reports that either did not analyze disseminated disease, or did not identify differences in fungal burden, leukocyte recruitment to affected sites, or mortality. Our data suggest that Card9 is required for the control of fungal burden in the lungs as well as dissemination and replication in the brain through various mechanisms that include the expression of cytokines and chemokines, as well as the recruitment of myeloid and lymphoid cell subsets. The consequence of these broad Card9-dependent defects is universal mortality from an otherwise non-lethal infection model.

Our study has several limitations that should be pointed out. First, we used a single inbred genetic background and one well-characterized *C. neoformans* isolate for our studies; accordingly, these results may not be generalizable to other inbred mice or cryptococcal strains. It would be informative to analyze the role of Card9 on host resistance to a highly virulent strain such as *C. neoformans* H99 or *C. gattii* VGII on the Balb/c or other genetic backgrounds. Second, while a marked increase in brain fungal burden was identified in Card9^em1Sq^ mice, we did not characterize the immune response in the central nervous system and cannot propose potential mechanisms of susceptibility as we have managed for infection of the lungs. Third, we did not determine which CLRs activate Card9 in response to intratracheal infection with *C. neoformans* 52D.

## 5. Conclusions

Balb/c mice with mutant Card9 are highly susceptible to progressive and disseminated infection with *C. neoformans* 52D. Card9 controls fungal replication in the lungs by regulating the expression of inflammatory, Th1-, and Th17-type cytokines, as well as a broad range of chemokines. The altered inflammatory environment is associated with the reduced recruitment of monocytes, dendritic cells, and neutrophils, as well as CD4^+^ and Ifn-γ^+^ CD4^+^ T cells. The resulting histological pattern is consistent with Th2-associated inflammatory/immune response. Finally, following intratracheal infection with *C. neoformans* 52D, Card9 mutant mice have a greater incidence of fungal dissemination and significantly higher fungal burden in the brain that results in 100% mortality. While human Card9 mutations have been specifically associated with susceptibility to *C. albicans* infection of the brain, the experimental data in this report raise the possibility that mutations in the Card9 gene may also be a risk factor for severe and/or disseminated cryptococcal disease.

## Figures and Tables

**Figure 1 jof-10-00434-f001:**
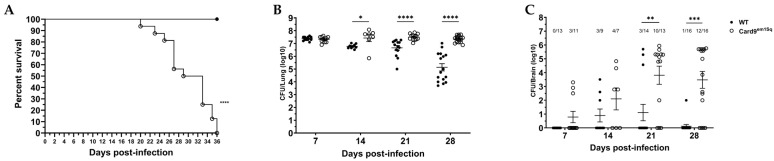
Card9 signaling is required for survival and control of fungal burden after infection with *Cryptococcus neoformans* 52D. Wild-type (WT) and Card9^em1Sq^ mice were infected intratracheally with 10^4^ CFU of *C. neoformans* strain 52D. (**A**) Mice were observed for up to 36 days for survival analysis (n = 16 mice/strain, using a log-rank test). (**B**,**C**) Fungal burden in the lung and brain at serial time intervals was determined by plating tissue homogenates on Sabouraud dextrose agar. CFU data are shown as mean ± SEM and combine at least two independent experiments (n = 9–16 mice /strain/time point). * *p* ≤ 0.05, ** *p* ≤ 0.01, *** *p* ≤ 0.001, and **** *p* ≤ 0.0001.

**Figure 2 jof-10-00434-f002:**
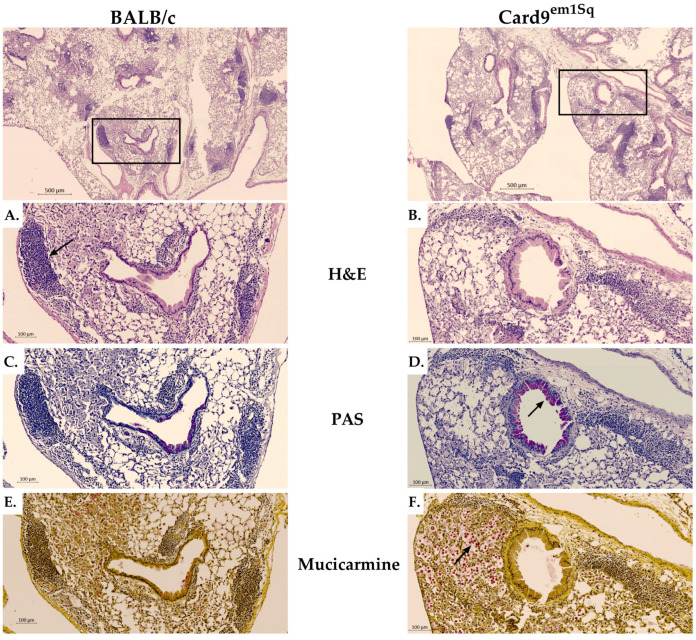
Histological analysis of *C. neoformans* pulmonary infection. Wild-type (WT) and Card9^em1Sq^ mice were infected intratracheally with 10^4^ CFU of *C. neoformans* 52D. Lungs were harvested at day 21 post-infection; perfused with phosphate-buffered saline; embedded in paraffin; and stained with (**A**,**B**) hematoxylin–eosin (H&E), (**C**,**D**) periodic acid–Schiff (PAS) or (**E**,**F**) mucicarmine. Rectangular boxes correspond to magnified regions (Balb/c; (**A**,**C**,**E**) and Card9^em1Sq^; (**B**,**D**,**F**)). Arrows identify findings that are detailed in the text of the manuscript.

**Figure 3 jof-10-00434-f003:**
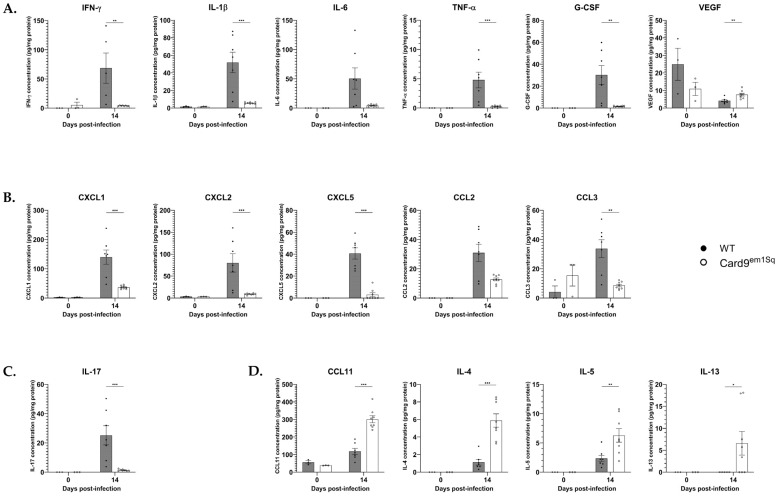
Pulmonary inflammatory mediator expression after *C. neoformans* infection. Wild-type (WT) and Card9^em1Sq^ whole lung proteins were collected at 14 days post infection with 10^4^ CFU of *C. neoformans* strain 52D. Milliplex and ELISA (IFN-γ) were performed to determine the level of (**A**) pro-inflammatory immune mediators, (**B**) chemokines, (**C**) Type 3 and (**D**) Type 2 cytokines. Mediators with levels below detectable levels are not shown. Data is shown as mean ± SEM. * *p* ≤ 0.05, ** *p* ≤ 0.01, and *** *p* ≤ 0.001.

**Figure 4 jof-10-00434-f004:**
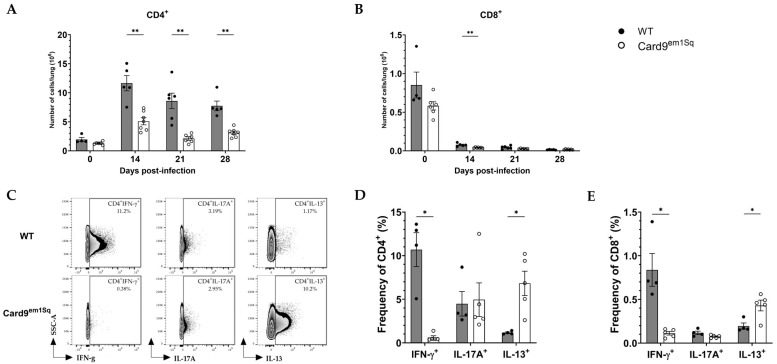
Pulmonary T-lymphocyte recruitment following *C. neoformans* infection. Lungs of Card9^em1Sq^ mice display fewer CD4^+^ and CD8^+^ T lymphocytes during the adaptive phase of immunity after *Cryptococcus neoformans* 52D infection. Total number of (**A**) CD3^+^CD4^+^ T lymphocytes and (**B**) CD3^+^CD8^+^ T lymphocytes in the lungs at 0, 14, 21, and 28 days post-infection. (**C**) Representative flow plots and (**D**) percentages of IFN-γ^+^CD4^+^, IL-17A^+^CD4^+^ and IL-13^+^CD4^+^ T cells. (**E**) percentages of IFN-γ^+^CD8^+^, IL-17A^+^CD8^+^ and IL-13^+^CD8^+^ T cells. Data are shown as mean ± SEM (n = 5–7 mice/strain/time point). * *p* ≤ 0.05 and ** *p* ≤ 0.01.

**Figure 5 jof-10-00434-f005:**
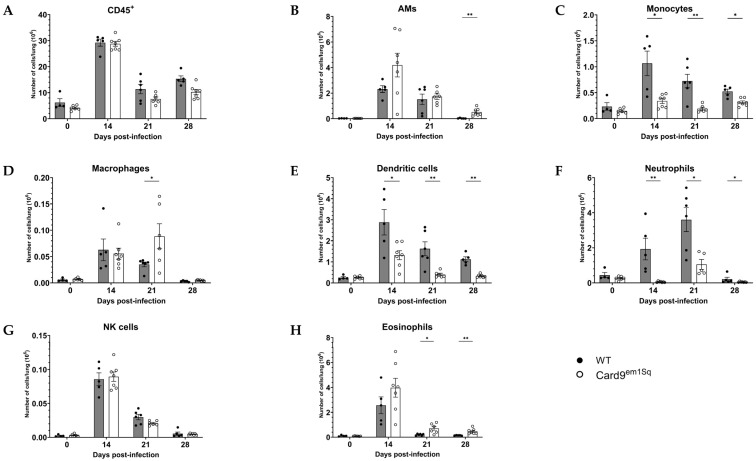
Pulmonary myeloid cell recruitment following *C. neoformans* infection. Lungs of Card9^em1Sq^ mice have decreased monocytes, dendritic cells, and neutrophil recruitment and increased alveolar macrophages and eosinophil recruitment to the lungs after *Cryptococcus neoformans* 52D infection. (**A**) Absolute numbers of total CD45^+^ cells in the lungs at 0,14, 21 and 28 days post-infection. Total number of (**B**) alveolar macrophages (CD45^+^CD11c^+^F4/80^+^SiglecF^+^), (**C**) monocytes (CD45^+^SiglecF^−^CD11b^+^Ly-6G^−^Ly-6C^hi^), (**D**) macrophages (CD45^+^SiglecF^−^CD11b^+^Ly-6G^−^Ly-6C^lo-int^CD11c^−^ F4/80^+^) (**E**) dendritic cells (CD45^+^SiglecF^−^CD11b^+^Ly-6G^−^Ly-6C^lo-int^CD11c^+^) (**F**) neutrophils (CD45^+^SiglecF^−^CD11b^+^Ly-6G^+^Ly-6C^hi^) (**G**) NK cells (CD45^+^SiglecF^−^CD11b^−^NK1.1^+^), and (**H**) eosinophils (CD45^+^CD11c^−^F4/80^int-hi^) in the lungs at 14, 21 and 28 days post-infection. Data are shown as mean ± SEM (n = 5–7 mice/strain/time point). * *p* ≤ 0.05 and ** *p* ≤ 0.01.

## Data Availability

The original contributions presented in the study are included in the article/Appendix A, further inquiries can be directed to the corresponding author.

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
