# Peer review of "Card9 Broadly Regulates Host Immunity against Experimental Pulmonary Cryptococcus neoformans 52D Infection"

_jof, 2024, doi:10.3390/jof10060434_

Round 1

Reviewer 1 Report

Although the role of CARD9 in antifungal immunity is critical, its specific function in anti-cryptococcal immunity has remained unclear/uncertain, primarily due to the limitations associated with the susceptibility of the commonly used C57BL/6J mice. In this paper, Angers at. al. successfully developed a new genetically modified mouse strain on the BALB/c background and effectively demonstrated the significance of this pathway in combating cryptococcal infection. Without CARD9, the mice become highly susceptible and are unable to control fungal growth. The observed decrease in Th1 cytokines and CD4 T cells is notable. It is a nice study well designed and executed.

While the involvement of CARD9 in the Th17/neutrophil axis is well-documented, this paper importantly highlights a previously underappreciated link between CARD9 and antifungal Th1 responses. The connection between protective anticryptococcal Th1 responses and CARD9 should be further explored in the discussion, including potential cells (such as macrophages or dendritic cells) expressing CARD9 and the mechanisms through which CARD9 promote robust antifungal Th1 immunity. This aspect might be beyond the scope of the current study but warrants mention.

Figure 2 could be enlarged, adding arrows to better illustrate their findings.

Reviewer 2 Report

In this report, the authors constructed Card9em1Sq mutant mice that lack exon 2 of the Card9 gene and intratracheally infected with C. neoformans 52D to investigate the effect of in immunity effects Card9 against C. neoformans. They found that Card9 broadly regulates the host inflammatory and immune response to experimental pulmonary infection with C. neoformans. The topic is important, and the results is well structured and straightforward. I have concerns needed to be addressed by the authors.

1.The number of keywords is too many, and should be decreased.

2.The authors should add some validation results of Card9em1Sq mutant mice.

3.The title of Figure 2, 3, 4, 5 should be more specifically.

4.In the title of Figure 2, the Latin scientific name of the species needs to be in italics.

5.The schematic diagram of possible mechanism of Card9 should be added.

Round 2

Reviewer 2 Report

All my concerns have been addressed by authors.

All my concerns have been addressed by authors.